# Two-Dimensional High-Performance Thin-Layer Chromatography with Bioautography for Distinguishing Angelicae Dahuricae Radix Varieties: Chemical Fingerprinting and Antioxidant Profiling

**DOI:** 10.3390/plants13101348

**Published:** 2024-05-13

**Authors:** Sejin Ku, Geonha Park, Young Pyo Jang

**Affiliations:** 1Department of Biomedical and Pharmaceutical Sciences, Graduate School, Kyung Hee University, Seoul 02447, Republic of Korea; zbxl0910@naver.com; 2Division of Pharmacognosy, College of Pharmacy, Kyung Hee University, Seoul 02447, Republic of Korea; ginapark0326@khu.ac.kr; 3Department of Integrated Drug Development and Natural Products, Graduate School, Kyung Hee University, Seoul 02447, Republic of Korea

**Keywords:** Angelicae Dahuricae Radix, *Angelica dahurica*, *Angelica dahurica* var. *formosana*, 2D-HPTLC, DPPH bioautography, HPTLC-DPPH, authentication

## Abstract

Angelicae Dahuricae Radix (ADR) holds a prominent place in traditional medicine for its remarkable antioxidative, anti-allergic, and antiproliferative capabilities. Recognized within the *Korean Pharmacopoeia* (KP 12th), *Angelica dahurica* (Hoffm.) Benth. and Hook.f. ex Franch. and Sav. (AD) and *Angelica dahurica* var. *formosana* (H. Boissieu) Yen (ADF) serve as the botanical origins for ADR. Differentiating these two varieties is crucial for the formulation and quality control of botanical drugs, as they are categorized under the same medicinal label. This research utilized two-dimensional high-performance thin-layer chromatography (2D-HPTLC) to effectively distinguish AD from ADF. Additionally, a quantitative analysis reveals significant differences in the concentrations of key active constituents such as oxypeucedanin, imperatorin, and isoimperatorin, with AD showing higher total coumarin levels. We further enhanced our investigative depth by incorporating a DPPH bioautography, which confirmed known antioxidant coumarins and unearthed previously undetected antioxidant profiles, including byakangelicin, byakangelicol, falcarindiol in both AD and ADF, and notably, 2-linoleoyl glycerol detected only in AD as an antioxidant spot. This comprehensive approach affords a valuable tool set for botanical drug development, emphasizing the critical need for accurate source plant identification and differentiation in ensuring the efficacy and safety of herbal medicine products.

## 1. Introduction

Angelicae Dahuricae Radix (ADR) has traditionally been utilized across East Asia—Korea, China, Japan, and Taiwan—and holds a significant position in traditional medicine, marked by its first mention in the *Shennong Bencaojing* [1], the earliest recorded text on herbal remedies. ADR is a key component in traditional formulations such as Gumiganghwal-tang, an herbal mixture extract used primarily for treating colds [2]. Gumiganghwal-tang consists of Angelicae Dahuricae Radix (3.6 g), Osterici seu Notopterygii Radix et Rhizoma (4.5 g), Saposhnikoviae Radix (4.5 g), Cnidii Rhizoma (3.6 g), Atractylodis Rhizoma (3.6 g), Scutellariae Radix (3.6 g), Fresh Rehmanniae Radix (3.6 g), Asiasari Radix et Rhizoma (1.5 g), and Glycyrrhizae Radix et Rhizoma (1.5 g), which work synergistically to relieve cold symptoms and support immune function. Additionally, ADR is prominently used in treatments for allergic rhinitis [3], as well as in external applications for skin whitening [4] and chloasma treatment [5,6]. The pharmacological versatility of ADR is largely attributed to its rich phytochemical composition, dominated by furanocoumarins such as imperatorin, isoimperatorin, and oxypeucedanin. These compounds are known for their antioxidative [7], acetylcholinesterase-inhibitory [8], hepatoprotective [9], anti-allergic [10], and antiproliferative [11] effects, which collectively highlight the therapeutic breadth of ADR in both traditional and modern medical practices. In addition to furanocoumarins, ADR contains a complex blend of essential oils, organic acids, and polysaccharides, which further contribute to its wide-ranging biological activities and underscore its therapeutic potential.

Within the framework of leveraging ADR for its varied therapeutic and activity-based applications, conventional practices have historically overlooked the distinction between the two botanical origins, *Angelica dahurica* (Hoffm.) Benth. and Hook.f. ex Franch. and Sav. (AD) and *Angelica dahurica* var. *formosana* (H. Boissieu) Yen (ADF), which are recognized origins of ADR in the *Korean Pharmacopoeia* (KP) [12], the *Pharmacopoeia of the People’s Republic of China* (ChP) [13], and the *Taiwan Herbal Pharmacopeia* (THP) [14]. Interestingly, the *Japanese Pharmacopoeia* (JP) [15] and the *Pharmacopoeia of the Democratic People’s Republic of Korea* (DPRKP) [16] only authorize the use of AD, not ADF. The standards specified in these pharmacopoeias vary by country, and differing botanical origins can lead to confusion in distribution and usage. This underscores the critical need for precise identification and chemical comparison of these two sources to ensure their effective therapeutic application and adherence to regulatory compliance.

To address these challenges, high-performance thin-layer chromatography (HPTLC) has emerged as a pivotal tool. HPTLC offers several advantages over conventional methods, including lower operational costs, faster analysis times, and the capability to analyze multiple samples simultaneously without extensive sample preparation [17]. The use of HPTLC in the authentication of AD and ADF is particularly valuable due to its effectiveness in resolving complex mixtures and providing clear, reproducible profiles ideal for comparative analysis [18].

However, AD and ADF being variants with closely similar morphological characteristics and potentially comparable compositions, differentiation of these two species based solely on morphological tests proves challenging. Given that AD is primarily grown in Korea [19] and ADF in China [20], geographical and cultivation conditions are expected to induce variations in their chemical constituents. These anticipated differences in phytochemical profiles, resulting from distinct environmental factors, underscore the necessity of conducting detailed chemical analyses to accurately distinguish between the closely related botanicals.

Previous research has explored various analytical techniques, such as HPLC-Q/TOF-MS [21], GC-MS [22], and DNA analysis [23], to differentiate AD from ADF. For instance, Shi et al., utilized HPLC-Q/TOF-MS to chemically compare and discriminate between AD and ADF, achieving quantitative analysis of multiple components with a single marker, which offers a high-resolution insight into the phytochemical profiles of these plants [21]. Hu et al., applied GC-MS to identify the aroma-active compounds in the root of ADR, providing essential data on volatile components that contribute to its sensory characteristics [22]. Additionally, Noh et al., employed a DNA-based multiplex SCAR assay to authenticate ADR, which, unlike the chromatographic methods, offers a genetic approach to ensure species-specific identification, highlighting its utility in overcoming limitations posed by phenotypic similarities [23]. However, there has been a notable absence of attempts employing high-performance thin-layer chromatography (HPTLC), despite its advantage of providing cost-effective, quick, parallel, easy-to-use analyses. This study, therefore, seeks to fill this gap by employing two-dimensional high-performance thin-layer chromatography (2D-HPTLC) to obtain comprehensive profiles of AD and ADF, aiming for the complete separation of key coumarins for content evaluation. Additionally, we compared the two species using bioautography profiling to confirm antioxidant activity within ADR, enhancing our understanding of their phytochemical differences and potential health benefits. The accurate distinction between AD and ADF is crucial for their application in the botanical drug industry, necessitating the establishment of distinct chemical profiles for each species.

## 2. Results

The TLC identification tests, as outlined by the pharmacopoeias of East Asia, were assessed and compared across the four countries, focusing on variations in extraction methods and analytical conditions, as depicted in Appendix A. Given the current pharmacopoeial identification tests, which are not designed to differentiate between the two species, AD and ADF were indistinguishable based on existing criteria. This evaluation led to the simplification of the extraction method, optimizing it for efficiency.

### 2.1. HPTLC Fingerprints

#### 2.1.1. One-Dimensional HPTLC

Under the development conditions specified by the KP using *n*-hexane and ethyl acetate (2:1, *v/v*), the analysis at UV 254 nm through 1D-HPTLC (Figure 1a) revealed single spots for isoimperatorin (*R*_F_ 0.44), imperatorin (*R*_F_ 0.27), and oxypeucedanin (*R*_F_ 0.21). However, upon examination at UV 365 nm, each of these compounds was observed to overlap with at least one additional spot, suggesting the presence of co-eluting compounds. Xanthotoxin and suberosin, presumed to be the overlapping compounds, shared matching *R*_F_ values with isoimperatorin and oxypeucedanin, respectively, confirming their co-location on the chromatographic fingerprint (Figure 1b).

#### 2.1.2. Two-Dimensional HPTLC

To surmount the limitations of 1D-HPTLC, the implementation of 2D-HPTLC enabled the separation of compounds that appeared indistinguishable in 1D due to identical *R*_F_ values, into discrete entities. Consequently, this approach significantly enhanced the capacity to identify a broader spectrum of compounds, as illustrated in Figure 2.

In the analysis of 20 samples using 2D-HPTLC, thymol was co-spotted on each plate as an internal standard to ensure system suitability across all plates. The development process positioned thymol at the coordinates (x = 20, y = 20), facilitating a standardized reference point for the evaluation. This approach allowed for the precise determination of each spot’s relative location on the plates, anchored by the predetermined position of thymol observed at a wavelength of 254 nm, where its visibility is optimized.

In the two-dimensional analysis, the bands observed at *R*_F_ 0.44 in 1D-HPTLC were discerned as distinct spots for isoimperatorin (x = 16.5, y = 16.0) [24] and suberosin (x = 14.5, y = 15.5) [21], while the bands near *R*_F_ 0.27 separated into three spots located at (x = 13.0, y = 9.5), (x = 10.5, y = 9.5), and (x = 6.5, y = 9.0). Among these, the mass-to-charge (*m/z*) ratios for imperatorin (x = 13.0, y = 9.5) and angelol E (x = 6.5, y = 9.0) were successfully identified using the MS interface [25]. However, the MS results at the (x = 10.5, y = 9.5) position, with *m*/*z* values of 351.12022 and 579.16912, indicated that this spot could not be attributed to a single substance. The bands located around *R*_F_ 0.21 underwent separation into two or more distinct entities, among which the spots for oxypeucedanin (x = 7.5, y = 7.0) and xanthotoxin (x = 10.5, y = 7.5) were successfully identified [25] (Figure 2).

### 2.2. Quantitative Analysis of Coumarins

Through the utilization of 2D-HPTLC, we achieved the complete separation of the spots associated with the major coumarins: oxypeucedanin, imperatorin, and isoimperatorin. The content of each isolated spot was subsequently quantified using a TLC scanner. By conducting a TLC scan at 366 nm, the peak areas of each spot were quantified in both AD and ADF samples (Figure 3a). The y-axis represents the average area of the respective substances across 10 samples each from AD and ADF. A comparison of the mean values for each substance revealed a significant difference in the content of oxypeucedanin, with a *p*-value of less than 0.001, underscoring a notable discrepancy in content between the two species. The average concentrations of the dried raw materials, as determined from the calibration curve of each standard compound, were as follows: for oxypeucedanin, 0.668 mg/g in ADF and 1.383 mg/g in AD; for imperatorin, 0.048 mg/g in ADF and 0.035 mg/g in AD; and for isoimperatorin, 0.216 mg/g in ADF and 0.284 mg/g in AD. Additionally, an analysis considering the total content of coumarins, which includes the three substances, demonstrated that AD possesses higher concentrations of these compounds (Figure 3b). This outcome aligned with previously reported results that compared the content of compounds in AD and ADF using HPLC-Q/TOF-MS [21]. Given these results, it could be speculated that among the two botanical origins of ADR, AD exhibits superior quality compared to ADF.

### 2.3. DPPH Bioautography on 2D-HPTLC

DPPH bioautography allows for the visualization of antioxidant activity, where active compounds reduce the purple DPPH reagent to a colorless or yellow form. This reduction can be observed as clear spots against a purple background on the TLC plate. Owing to its antioxidative properties, thymol, defined at coordinates (x = 20.0, y = 20.0), underwent a reaction with the DPPH solution, thereby validating the functionality of TLC–bioautography as an effective analytical method (Figure 4a,b). Known as key coumarin components of ADR, isoimperatorin, imperatorin, and oxypeucedanin, highlighted with black circles among the spots, demonstrated antioxidant properties in both AD and ADF samples. Additionally, through the DPPH assay, we identified three new spots, highlighted with a green circle and some yellow circles, that exhibit antioxidant activity, which were difficult to detect prior to the DPPH analysis. Among these, two substances, excluding the spot located at (x = 29.0, y = 27.0), were identified as 2-linoleoyl glycerol and falcarindiol (Figure 4e), with 2-linoleoyl glycerol exclusively observed in AD samples. Notably, byakangelicin and byakangelicol, marked in yellow and also observable under 366 nm in the 2D-HPTLC analysis, indicate the existence of additional antioxidative coumarins within ADR, further expanding its antioxidative compound profile.

In Figure 4c,d are displayed the densitometric scanning results at 517 nm from a TLC scanner, where the absorbance corresponds to the intensity of the purple coloration that develops upon reacting with DPPH. This color intensity is a direct reflection of the presence and relative concentration of antioxidants, with lower absorbance indicating a greater quantity of antioxidant compounds. The spectra revealed that the absorbance levels for oxypeucedanin and byakangelicol are significantly lower in AD compared to ADF. Moreover, a peak activity unique to AD around an *R*_F_ of 0.1 confirms the exclusive presence of 2-linoleoyl glycerol. These observations demonstrated a stronger antioxidant capacity in AD, evidenced by the overall higher absorbance peaks. Such differential absorbance not only emphasized the varied antioxidant capacities but also served as a qualitative distinction between the two variants.

### 2.4. Identification of the Compounds from 2D-HPTLC and DPPH Bioautography

Table 1 presents the compounds identifiable in the chemical profiles of AD and ADF, as established through 2D-HPTLC combined with bioautography. Every substance was pinpointed via MS analysis and a literature review using the MS interface. Furthermore, validation for nine of these substances, with the exception of angelol E, was achieved by acquiring standard compounds and verifying their coordinates under identical 2D-HPTLC analysis conditions.

## 3. Discussion

HPTLC offers several advantages allowing for multiple-sample analysis under identical conditions and facilitating comparative studies and reducing variability. The technique’s capability to separate a wide range of compounds within a short time frame, coupled with low operational costs and minimal sample preparation, underscores its utility in rapid screening and detailed phytochemical profiling. Two-dimensional HPTLC offers significant advantages over 1D-HPTLC, particularly in resolving complex mixtures where components might overlap in a single dimension The choice of 2D-HPTLC over the use of multiple one-dimensional runs with different solvent systems stems from its ability to provide a comprehensive separation profile. TLC–bioautography has been suggested as an efficient technique for simultaneously observing the activity of all substances within an extract when applied to a single plate. The results obtained from the DPPH assay with 2D-HPTLC provided a compelling insight into the antioxidant profiles of AD and ADF (Figure 5). DPPH is non-specific and can react with a broader range of antioxidant compounds, providing a more comprehensive assessment of the antioxidant capacity of a sample. Additionally, the visual and quantitative results obtained through DPPH are generally more reproducible and can be directly related to the radical scavenging activity of the compounds, which is crucial in antioxidant studies. The reaction of thymol with the DPPH solution, validated at coordinates (x = 20.0, y = 20.0), confirms the utility of TLC–bioautography as a robust analytical method.

In our comparative analysis of AD and ADF, several key differences and similarities in their chemical profiles were identified, both qualitatively and quantitatively. Qualitatively, both species exhibited a similar range of bioactive compounds, particularly the presence of major coumarins such as imperatorin, isoimperatorin, and oxypeucedanin. However, quantitatively, significant variations were observed. AD consistently showed higher concentrations of these coumarins compared to ADF, as evidenced by the densitometric analysis in our HPTLC studies. This suggests that while the two variants share a common set of active components, the concentration of these compounds can vary substantially, likely influenced by genetic and environmental factors specific to each variant.

The additional finding of three new spots, previously undetected before the DPPH assay, introduces novel compounds of interest within the ADR spectrum. Notably, the differentiation of AD from ADF through the exclusive presence of 2-linoleoyl glycerol in AD, as delineated by a unique green circle, presents a significant marker for botanical distinction. This particular discovery, alongside the higher overall absorbance levels indicating superior antioxidant activity in AD, suggests that AD may offer enhanced pharmacological benefits compared to ADF. Furthermore, the identification of byakangelicin, byakangelicol, and falcarindiol, excluding the spot at (x = 29.0, y = 27.0), enriches the compound library of ADR and highlights the complexity of its antioxidant properties.

This study’s findings, particularly the superior antioxidant capacity observed in AD samples, suggest the potential for differential applications in medicinal formulations based on the specific ADR variety. The delineation of antioxidant profiles between AD and ADF could inform future research directions, focusing on the isolation, characterization, and pharmacological testing of these newly identified compounds. Moreover, the variability in antioxidant activity across AD and ADF emphasizes the importance of source verification and standardization in the production of herbal medicines, ensuring consistency in efficacy and safety for clinical use. The 2D-HPTLC-DPPH profiles generated in this study are anticipated to serve as a foundational dataset for natural product research. This dataset will enable further investigations into the antioxidant properties of these compounds and enhance the broader understanding of their potential therapeutic benefits.

## 4. Materials and Methods

### 4.1. Plant Materials

A total of 20 batches of ADRs, including 10 batches for each species, were acquired from the local market for analysis. The authenticity of the collected samples was verified by matching their HPTLC profiles with the reference medicinal plant materials provided by the Ministry of Food and Drug Safety (MFDS), with lot numbers ANDA2009 for AD and ANFO2009 for ADF. All samples were ground to a fine powder and passed through an 850 μm sieve. Subsequently, 1.0 g of the powdered sample was combined with 5 mL of methanol. This mixture was shaken for 10 min at 100 rpm at ambient temperature. After shaking, the mixture was centrifuged at 2000 rpm for 1 min. The supernatant obtained was then filtered through a 0.45 μm polytetrafluorethylene (PTFE) syringe filter (Whatman, Marlborough, MA, USA) to prepare the test solution.

### 4.2. Chemicals and Reagents

Methanol (99.8%), *n*-hexane (95.0%), ethyl acetate (99.5%), and magnesium chloride hexahydrate (98.0%) were obtained from Duksan (Ansan, Republic of Korea), formic acid (99.0%) from Daejung Chemicals & Metals (Siheung, Republic of Korea), and toluene (99.5%) from Junsei Chemical Co., Ltd. (Tokyo, Japan). Water (HPLC grade), methanol (HPLC grade), and 2,2-diphenyl-1-picrylhydrazyl (DPPH) were obtained from Thermo Fisher Scientific (Waltham, MA, USA). Standard compounds, oxypeucedanin (98.3%, CFN90350-CFS202201), imperatorin (98.4%, CFN98758-CFS202301), isoimperatorin (98.1%, CFN99107-CFS202201), byakangelicol (99.1%, CFN98167-CFS202301), byakangelicin (99.4%, CFN98152-CFS202201), suberosin (98.8%, CFN98985-CFS202201), and falcarindiol (99.4%, CFN98220-CFS202301) were supplied by Chemfaces (Wuhan, China), xanthotoxin (98.0%, LOT AS01) from TCI (Tokyo, Japan), 2-linoleoyl glycerol (95.0%) from Glpbio (Montclair, NJ, USA), and thymol (98.5%) from Merck (Darmstadt, Germany). Individual standards were dissolved in methanol to create solution stocks with a concentration of 1.0 mg/mL.

### 4.3. HPTLC Analysis

#### 4.3.1. Equipment

HPTLC examinations utilized 20 × 10 cm or 10 × 10 cm silica gel 60 F_254_ glass plates (Merck, Darmstadt, Germany) across several HPTLC setups provided by CAMAG (Muttenz, Switzerland). These setups included a Linomat 5 semiauto-applicator, an Automatic Developing Chamber (ADC 2), a Visualizer 2, a TLC Scanner 4, and *visionCATS* software, version 3.1. *visionCATS* software was employed for data analysis, controlling the system, and performing quantification. For sample application, a 100 μL syringe (Hamilton, Bonaduz, Switzerland) was employed utilizing nitrogen gas (Sinyang Sanso, Seoul, Republic of Korea).

#### 4.3.2. Analytical Conditions for 1D-HPTLC

For 1D-HPTLC, 10 samples from each species were applied on 20 × 10 cm individual plates. Each solution was dispensed in volumes of 2 μL as 8 mm bands, 11.4 mm apart and 10 mm from the plate’s bottom edge, at a delivery speed of 150 nL/s. The first band was positioned 20 mm from the plate’s side edge. The development was standardized by conditioning the layer to 33% relative humidity using a saturated aqueous solution of magnesium chloride for 10 min. Following this conditioning, the plates were developed in a chamber pre-saturated for 20 min with a saturation pad, up to a distance of 80 mm from the plate’s bottom edge. The plates were then dried for 5 min. The developing solvent was composed of *n*-hexane and ethyl acetate, mixed in a 2:1 volume ratio, to achieve a total volume of 45 mL. Out of the prepared solvent, 10 mL was utilized for the development process, while 25 mL was allocated for chamber saturation. After developing, the 1D-HPTLC chromatograms were obtained under UV 254 nm and 366 nm.

#### 4.3.3. Analytical Conditions for 2D-HPTLC

For 2D-HPTLC, each sample was applied on 10 × 10 cm individual plates. The sample application point on the plate was positioned 10 mm from its lower edge and 10 from the left side, with each sample being injected in a volume of 3 μL to form a band of 2 mm width, at a delivery speed of 150 mL/s. This narrow width of the band ensures a uniform starting point for the second phase of 2D analysis, enhancing consistency across runs. Additionally, 1 μL of thymol solution, acting as an internal standard, was applied at the same location. This approach supported the system suitability test by enabling the evaluation of the positions of various substances relative to thymol. The chromatographic process is bifurcated into two phases. Initially, the procedure mirrors that of 1D chromatography, described above. Following this initial phase, the plate undergoes a 90-degree rotation to commence the second development stage, adopting an orthogonal strategy to the first. The second phase employs a mobile phase mixture of toluene, ethyl acetate, and formic acid, mixed in a 9:1:0.1 volume ratio, aggregating to a total of 50.5 mL. This setup maintains a development distance of 70 mm from the application point for both the 1D and subsequent 2D analysis stages, ensuring thorough and uniform chromatographic exploration. After developing, the 2D-HPTLC chromatograms were obtained under UV 254 nm for coordination and UV 366 nm for profiling.

For semi-quantification of coumarins, the developed plates were scanned at UV 366 nm using a TLC scanner. The developed section of the chromatographic plate, spanning from 1 cm to 8 cm, was segmented into 40 distinct tracks for detailed scanning. The quantification of individual coumarin compounds was conducted by precisely identifying the area corresponding to the *R*_F_ value of the major coumarins within each track. Comprehensive quantification of the total coumarins’ content was then accomplished by aggregating the areas of all detected spots at UV 366 nm. The parameters for the scan included a slit dimension of 5 × 0.2 mm, a scanning speed of 20 mm/s, and a data resolution of 100 μm per step. The scanning span covered a range from 6.5 mm to 83.5 mm, utilizing a mercury lamp and a K400 filter to facilitate detection.

### 4.4. DPPH Bioautography

The application of the TLC–bioautography method for verifying the presence of antioxidants through radical scavenging activity draws on established precedents in research [19,20,21,22,23,24]. The preparation of the DPPH solution involves dissolving DPPH in methanol, followed by sonication to achieve a 0.1% (*w*/*v*) concentration. This solution is prepared fresh and shielded from light exposure. During the assay, the developed HPTLC plate, positioned face-down, is briefly immersed in the DPPH solution for 10 s before being withdrawn. It is then allowed to stand at ambient temperature in a well-lit environment. After a period of 6 h, regions appearing lighter against the purple backdrop are identified, signifying areas of antioxidant activity. These regions are captured photographically under white light using a TLC visualizer and scanned at 517 nm using a TLC scanner with a deuterium and tungsten mode.

### 4.5. MS-Interface Analysis

For the elucidation of compounds corresponding to specific spots on the HPTLC plate, a JMS-T100TD Time of Flight (TOF) mass spectrometer (JEOL Ltd., Tokyo, Japan) with an electrospray ionization (ESI) source was employed to acquire mass spectrometry data. The CAMAG TLC-MS interface, facilitating the integration of chromatographic and MS analyses, was connected to a binary pump and the mass spectrometer. To accurately target chromatographic zones for HPTLC-MS investigation, these areas were delineated under UV 366 nm. This step ensured precise alignment of the HPTLC plate beneath the oval elution head (2 × 4 mm) of the TLC-MS interface, enabling the efficient elution of compounds into the mass spectrometer. The elution process utilized a flow rate of 0.3 mL/min with 80% methanol as the eluent. The MS operational parameters were as follows: a mass-to-charge ratio (*m*/*z*) range of 50 to 1500, with a scan interval of 0.5 s, and analysis conducted in positive ion mode. Detailed settings included an orifice 1 temperature of 80 °C, a desolvating chamber temperature of 250 °C, detector voltage at 2000 V, ring lens voltage at 15 V, orifice 1 voltage at 40 V, orifice 2 voltage at 7 V, needle voltage at 2000 V, peak voltage at 1500 V, nebulizing nitrogen gas flow at 1 L/min, and desolvating nitrogen gas flow at 3 L/min. Calibration of the mass scale for accurate mass measurement was conducted using the YOKUDELNA solution (JEOL Ltd.) and data acquisition was performed using Mass Center software, version 1.3.7b (JEOL Ltd.)

## Figures and Tables

**Figure 1 plants-13-01348-f001:**
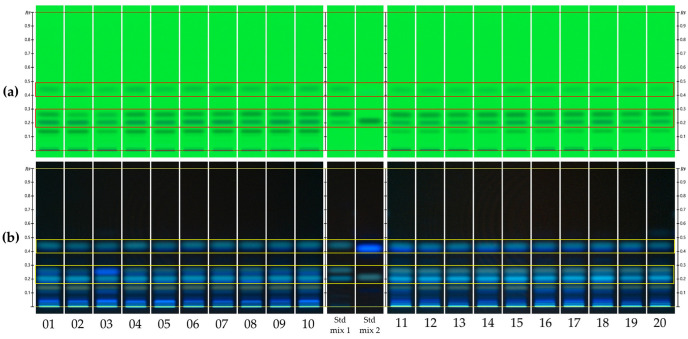
One-dimensional HPTLC fingerprints of AD (track 01–10) and ADF (track 11–20) samples at UV 254 nm (**a**) and UV 366 nm (**b**) with a development solvent system of hexane and ethyl acetate (2:1, *v/v*). Std mix 1: oxypeucedanin, imperatorin, and isoimperatorin (as *R*_F_ increasing); std mix 2: xanthotoxin, and suberosin (as *R*_F_ increasing).

**Figure 2 plants-13-01348-f002:**
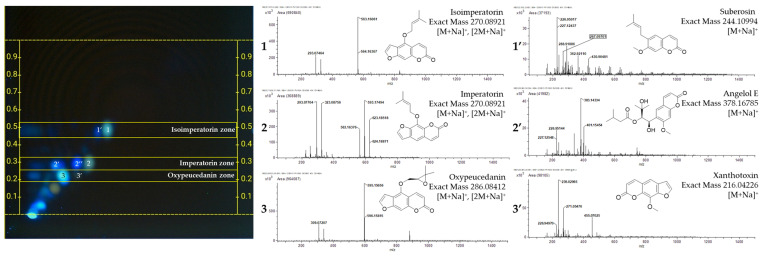
Representative 2D-HPTLC fingerprint and mass spectra of the major spots; 1: isoimperatorin, 1′: suberosin, 2: imperatorin, 2′: angelol E, 2′’: undefined, 3: oxypeucedanin, 3′: xanthotoxin.

**Figure 3 plants-13-01348-f003:**
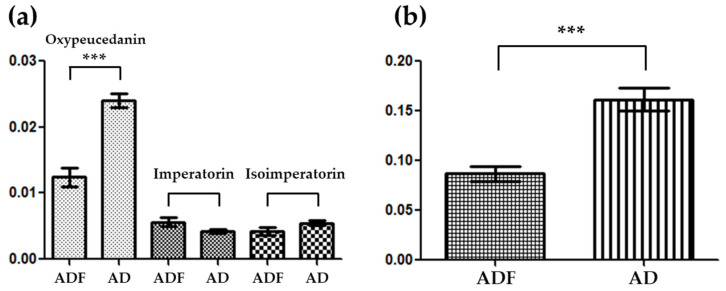
The comparative analysis of the contents of oxypeucedanin, imperatorin, and isoimperatorin in AD and ADF: (**a**) individual coumarins; (**b**) total coumarins. *p*-Values are significant according to the Student’s *t*-test. *** *p*
< 0.001.

**Figure 4 plants-13-01348-f004:**
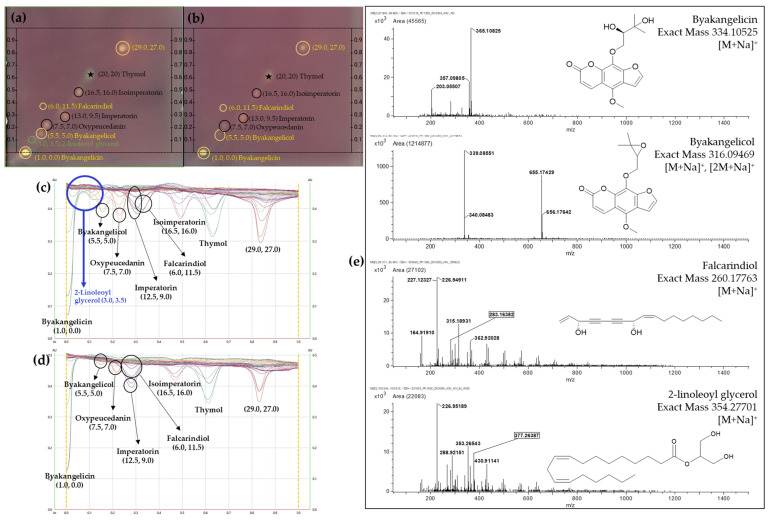
Representative DPPH bioautography fingerprints of AD (**a**) and ADF (**b**) under RT white light, spectra at 517 nm of AD (**c**) and ADF (**d**), and the MS spectra for new antioxidant spots (**e**).

**Figure 5 plants-13-01348-f005:**
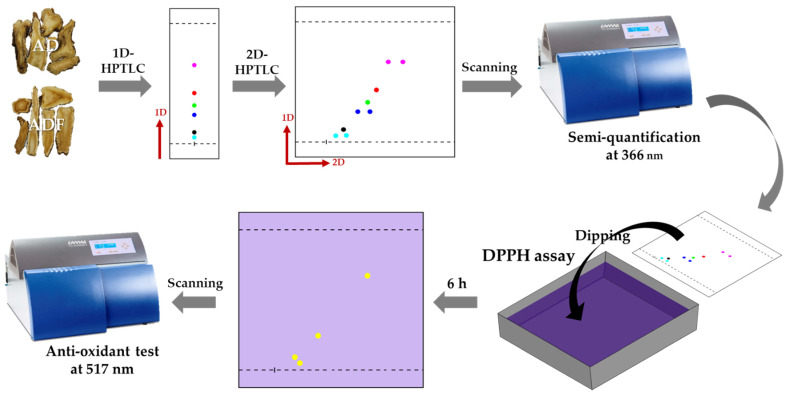
The overall work flow of the 2D-HPTLC with bioautography.

**Table 1 plants-13-01348-t001:** All identified compounds in the chemical profiles of AD and ADF.

No.	Position (x, y)	*m*/*z* ^1^	Quasi-Molecular Ion	Mass Difference (mu)	Molecular Formula	Identification	Ref
1	(16.5, 16.0)	293.07464	[M+Na]^+^	−4.34	C_16_H_14_O_4_	Isoimperatorin	[24] ^2^
1′	(14.5, 15.5)	267.09701	[M+Na]^+^	−2.71	C_15_H_16_O_3_	Suberosin	[21] ^2^
2	(13.0, 9.5)	293.07704	[M+Na]^+^	−1.93	C_16_H_14_O_4_	Imperatorin	[25] ^2^
2′	(6.5, 9.0)	401.15454	[M+Na]^+^	−3.08	C_20_H_26_O_7_	Angelol E	[25]
3	(7.5, 7.0)	309.07207	[M+Na]^+^	−1.82	C_16_H_14_O_5_	Oxypeucedanin	[25] ^2^
3′	(10.5, 7.5)	239.02965	[M+Na]^+^	−2.38	C_12_H_8_O_4_	Xanthotoxin	[25] ^2^
4	(6.0, 11.5)	283.16382	[M+Na]^+^	−3.58	C_17_H_24_O_2_	Falcarindiol	[26] ^2^
5	(5.5, 5.0)	339.08551	[M+Na]^+^	1.05	C_17_H_16_O_6_	Byakangelicol	[25] ^2^
6	(3.0, 3.5)	377.26387	[M+Na]^+^	−2.91	C_21_H_38_O_4_	2-Linoleoyl glycerol *	[27] ^2^
7	(1.0, 0.0)	357.09805	[M+Na]^+^	3.02	C_17_H_18_O_7_	Byakangelicin	[24] ^2^

^1^ Mass-to-charge ratio (*m/z*); ^2^ confirmed with standard compounds. * Only present in AD.

## Data Availability

The data presented in this study are available on request from the corresponding author.

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
