# Peer review of "Two-Dimensional High-Performance Thin-Layer Chromatography with Bioautography for Distinguishing Angelicae Dahuricae Radix Varieties: Chemical Fingerprinting and Antioxidant Profiling"

_plants, 2024, doi:10.3390/plants13101348_

Round 1
Reviewer 1 Report
Comments and Suggestions for Authors
Plants_2980554_Review
Overall Comment:
· This technical work provides comprehensive methodologies for authenticating a major plant sample using HPTLC. Additionally, it emphasizes the utilization of DPPH bioautography to ascertain the bioactive principles present in the plant samples. Overall, it presents an intriguing approach to plant sample analysis and authentication.
Supplementary Materials:
· Table S1. Scientific name must be italicized
· Indicate N.A. or not applicable for the developing solvent and developing condition under JP
· Figure S1. Label the two images as A and B and the description of the solvent system used must be imbedded in the figure title. Furthermore, light conditions must also be indicated for each images.
· Figure S1. Figure S1. The HPTLC chromatograms of different pharmacopoeias conditions. This needs to be clarified and specified.
· Figure S1. Include Track for each numbers
· Were the TLC fingerprints shown in Figure S1 derived from a literature? If yes, cite the source accordingly.
Title and Abstract:
· Scientific name in the title must be italicized.
· The title can be improved
· The abstract should follow the format of stating the aim or purpose of the study, followed by methodology, results, and conclusion
· In the results part of the abstract, indicate quantities if there was a quantification and indicate if what cmpounds should activity against the DPPH radical.
·
Keywords:
· Maximize the number of keywords. You can include Angelica dahurica var. formosana, Angelicae dahuricae, high-performance thin-layer chromatography, HPTLC-DPPH, authentication, etc.
Introduction:
· Since this paper delves into authentication, there is a need to enumerate the compounds that had previously been identified in AD and ADF and then indicate the instrumentation used to identify and quantify it. You can also highlight the advantages and disadvantages of each methods.
· HPTLC must also be included in the introduction, indicating its advantages over the conventional methods and indicate the extent of the use of HPTLC in the authentication of AD and ADF and then, indicate why your research and your data is unique and significant in the field.
Results
· Section 2.1.1.1. Should be 1D-HPTLC
· Figure 1. Add the solvent system used
· Section 2.1.1.1. A Question: Have you tried to optimize the solvent system to ensure that the marker compounds do not overlap with each other?
· Section 2.1.1.2. Should be 2D-HPTLC
· Figure 3. The description of Figure 3 is unclear. Revise to improve clarity. Also, include the level of significance and at which p-value.
· Figure 3. If possible, show the lower error line as well.
· Figure 4. Spell out RT
· Figure 2, 4, and Table 1. It would be helpful if the MS spectra and the tables would contain the compound structures.
· Figure 4, c and d. Please explain the chromatograms in images c and d.
· Section 2.4. The results of the DPPH bioautography have to be explained further and should be interpreted even from a qualitative point of view.
· Table 1. Since validation was performed, include at least the % recovery for each compound that were subjected to validation.
Discussion:
· The discussion should include more important aspects of the versatility of HPTLC
· Discuss why DPPH is a better derivatizing agent compared to vanillin sulfuric acid, natural product reagent, etc.
· Indicate the advantages of 2D vs 1D HPTLC in the analysis
· Discuss why 2D HPTLC was chosen as compared to the use of 2 or more solvent systems
· Discuss the validation procedures particularly, the recovery.
· Indicate the limitations and future directions of the study
Materials and Methods:
· Section 4.2 Should be Chemicals and Reagents
· Section 4.3.2 and Section 4.3.3. Indicate the pressure of nitrogen applied in the sample application. Also indicate that Vision Cat software was used for data analysis, controlling the system, and quantification.
· Section 4.3.2. Indicate the photodocumentations performed. 254 nm? 366 nm? Etc.
· Section 4.3.2. Did you use any reference standards in running the plates?
· Section 4.3.3. Indicate the concentration of thymol used
· Section 4.3.3. Indicate the photodocumentations performed.
· Section 4.3.3. For the quantification, specify the wavelength/s used in scanning to detect the compounds. Also, indicate the procedure on how the individual coumarins and the total coumarins were quantified.
· Section 4.4. Why is there a need to allow the plate to stand for 6 hours after being immersed in DPPH? Is 1 hr or 2 hrs not enough? Usually, DPPH degrades over time.
· Section 4.5. Indicate in the methodology if how the compounds were identified using the fragmentations. Were software or MS library used? If yes, describe them in detail in the procedure and then, provide an example of compound matching in the results section.
Conclusion:
· There is no conclusion.
Comments on the Quality of English Language
The paper requires thorough proofreading for grammar, spelling, and to improve cohesion of thoughts.
Reviewer 2 Report
Comments and Suggestions for Authors
The authors presented a paper on the 2D-HPTLC with Bioautography for Distinguishing Angelicae Dahuricae Radix Varieties: Chemical Fingerprinting and Antioxidant Profiling.
The topic is interesting and within the aims and scopes of the Journal.
Yet, some major implementations are necessary as reported below one by one:
- Line 16: Please write the authors’ names of the species here.
- Line 21: Avoid repetition of concepts.
- Line 28: In ensuring?
- Line 32: Please write something about the aspects and preparation of ADR.
- Please write something about this Gumiganghwaltang: what plants, what formulation?
- Lines 49-51: Why?
- Lines 55-57: Please explore these morphological differences.
- Why only coumarins for this study? The species are well known also to biosynthesize flavonoids, sterols, benzofurans and poly-acetylenes which may also be responsible for the biological properties associated to this species.
- In Table S1, you must specify the meaning of KP and the rest. I would move this table in the main text.
- Please discuss about Table S1 methods, too. Are they fine or not in your opinion in general?
- Line 86 on: Please write n-hexane with n in Italics.
- Line 87 on: Please write v/v in Italics.
- Figure 1 is almost the same as Figure S1. Erase the last one.
- Lines 111 on: What are those values in parentheses near the name of the compound in practice? Please report it.
- Line 113 on: m/z must be in Italics.
- And what about the identification of the rest of the spots? This would make your work more complete.
- I do not see any real value for quantification of these coumarins. In addition, quantified with respect to what? The dried extract or other?
- And the values of this DPPH activity for the two species are exactly?
- And what are the total differences and similarities between the two species in quality and quantity? Not very clear.
- What about the occurrence of these other compounds in the species? Are they already known in them?
- You must report the batch numbers of these twenty samples.
- And the original collection areas of the samples?
- Line 257: Concoction?
- You must always write v/v in Italics near all the concentration ratios.
- Line 270 on: w/v must be in Italics.
- The references are not reported in the list as exactly requested by the Journal.
Comments on the Quality of English LanguageReported previously
Round 2
Reviewer 1 Report
Comments and Suggestions for Authors
Dear Authors,
Thanks you for addressing my queries and suggestions. The paper is now of good quality and ready for publication.
Comments on the Quality of English Language
proof read for typographical and grammatical errors.
Author Response
Thanks for your thoughtful evaluation and suggestions.
Reviewer 2 Report
Comments and Suggestions for Authors
The authors presented a revised version of the paper I have previously reviewed.
Some improvements have been observed but major concerns remain:
- A complete list of Gumiganghwaltan plants is needed also in the proportions.
- You should write more correctly (x = 16.5, y = 16.0). Please modify all the rest accordingly.
- The quantification must be calculated with respect to the grams of dried extract. This is more precise and correct. In addition, the quantification must be precise to evaluate comparisons.
- I think giving exact values for DPPH is compulsory.
- The references are not still reported in the list as exactly requested by the Journal.
Author Response
- A complete list of Gumiganghwaltan plants is needed also in the proportions.
A: We have detailed the medicinal plants and their respective proportions that comprise Gumiganghwal-tang.
“Gumiganghwal-tang consists of Angelicae Dahuricae Radix (3.6 g), Osterici seu Notopterygii Radix et Rhizoma (4.5 g), Saposhnikoviae Radix (4.5 g), Cnidii Rhizoma (3.6 g), Atractylodis Rhizoma(3.6 g), Scutellariae Radix (3.6 g), Fresh Rehmanniae Radix (3.6 g), Asiasari Radix et Rhizoma (1.5 g), and Glycyrrhizae Radix et Rhizoma (1.5 g) which work synergistically to relieve cold symptoms and support immune function.”
- You should write more correctly (x = 16.5, y = 16.0). Please modify all the rest accordingly.
A: We have revised the descriptions of the positions of the spots for clarity.
“In the 2-dimensional analysis, the bands observed at RF 0.44 in 1D-HPTLC were discerned as distinct spots for isoimperatorin (x 16.5, y 16.0) [24] and suberosin (x 14.5, y 15.5) [21], while the bands near RF 0.27 separated into three spots located at (x 13.0, y 9.5), (x 10.5, y 9.5), and (x 6.5, y 9.0). Among these, the mass-to-charge (m/z) ratios for imperatorin (x 13.0, y 9.5) and angelol E (x 6.5, y 9.0) were successfully identified using the MS interface [25]. However, the MS results at the (x 10.5, y 9.5) position, with m/z values of 351.12022 and 579.16912, indicated that this spot could not be attributed to a single substance. The bands located around RF 0.21 underwent separation into two or more distinct entities, among which the spots for oxypeucedanin (x 7.5, y 7.0) and xanthotoxin (x 10.5, y 7.5) were successfully identified [25] (Figure 2).”
- The quantification must be calculated with respect to the grams of dried extract. This is more precise and correct. In addition, the quantification must be precise to evaluate comparisons.
A: As recommended, we have provided the average content values for oxypeucedanin, imperatorin, and isoimperatorin in mg/g for the plant power.
“The average concentrations of the dried raw materials, as determined from the calibration curve of each standard compound, were as follows: for oxypeucedanin, 0.668 mg/g in ADF and 1.383 mg/g in AD; for imperatorin, 0.048 mg/g in ADF and 0.035 mg/g in AD; and for isoimperatorin, 0.216 mg/g in ADF and 0.284 mg/g in AD.”
- I think giving exact values for DPPH is compulsory.
A: The DPPH assay employed in this study is considered to be a less precise model for determining accurate antioxidant values. Instead, we focused on using substances with known antioxidant activity, such as thymol as references for antioxidant potency. The relative antioxidant potency was assessed by comparing the degree of reduction in the negative peak of each component against that of thymol.
- The references are not still reported in the list as exactly requested by the Journal.Supplementary
A: We apologize for that. We’ve checked the journal’s reference formatting guidelines and updated the references accordingly.
Round 3
Reviewer 2 Report
Comments and Suggestions for Authors
All is fine now. The paper can be accepted